# Depression and associated factors among older people in Vietnam: Findings from a National Aging Survey

**Nam Truong Nguyen**[1]*, **Trang Nguyen**[1], **Thu Dai Bui**[1], **Long Thanh Giang**[2]

**1** Institute of Social and Medical Studies, Ha Noi, Vietnam, **2** National Economics University, Ha Noi, Vietnam

* ntnam@isms.org.vn

## Abstract

### Background

Depression is one of the most common mental health disorders among older people. Depressive symptoms are often overlooked and untreated in primary care settings. This study aims to assess the prevalence of depressive symptoms and associated factors among older people in Vietnam.

### Method

The study analyzed data from the Vietnam National Aging Survey (VNAS) conducted in 2022 with a nationally representative sample of 3,006 older people aged 60 and over in 12 provinces. The 15-item Geriatric Depression Scale (GDS-15) was used to assess depressive symptoms. Bivariate and multiple logistic regression analyses were used to explore the association between depressive symptoms and other related factors such as sociodemographic and economic characteristics, social support, health status, Activities of Daily Living (ADL) and Instrumental Activities of Daily Living (IADL) limitations, chronic diseases, cigarette smoking status, alcoholic drinking, and domestic violence.

### Results

The prevalence of depressive symptoms among older people was 20.2%. The associated factors that increase the odds of having depression among older people were female gender (OR = 2.21, 95% CI 1.34–3.62), living in rural areas (OR = 1.83, 95% CI 1.15–2.89), the poorest quintile (OR = 2.26, 95% CI 1.39–3.66), self-rated poor health (OR = 11.68, 95% CI 4.96–27.49), ADL limitations (OR = 2.12, 95% CI 1.51–2.99), IADL limitation (OR = 1.61 95% CI 1.16–2.25), and experiencing domestic violence in the last 12 months (OR = 6.66, 95% CI 4.00–11.05).

**Data Availability Statement:** All relevant data are within the paper and its Supporting information files designated as "S1 Dataset.dta".

**Funding:** The authors received no specific funding for this work.

**Competing interests:** The authors have declared that no competing interests exist.

## Conclusion

Depression symptoms were prevalent among older people in Vietnam. Depression screening for older people should be included in primary care settings for early identification and treatment of depression.

## Introduction

Economic development, improvements in living standards, and improved health/medical services have resulted in a rapidly increased life expectancy, and subsequently a rapidly aging population. The population aged 60 and older increased from 900 million (12% of the world population) in 2015 to 2 billion (22% of the world population) in 2050 [1]. Vietnam is one of the fastest-aging countries. The number of people aged 60 and over was 12.58 million which accounted for 12.8% of the total population in 2021 [2]. In 2050, 25% of the population will be over the age of 60 [3].

Gains in life expectancy are threatened by the growing burden of non-communicable diseases among older people. Depression is one of the most common mental health disorders among older people [1]. A systematic review and meta-analysis that included 42 studies with 57,486 older people in 23 countries show that the average expected prevalence of depression among older people was 31.74% [4]. Studies conducted in Vietnam show a similarly high prevalence of depression among older people, ranging from 17.2% to 39.6% [5–11].

Depression is associated with negative health outcomes including functional impairment, poor health status, decreased quality of life, increased health care utilization, and increased medical care cost. Depression is a risk factor for disability and mortality in older people [12–16]. Risk factors associated with depression among older people include older age, female gender, being unmarried, low income, lower education, loneness or living alone, low social support, presence of chronic diseases, poor health status, Activities of Daily Living (ADL) and Instrumental Activities of Daily Living (IADL) limitations [4, 17].

Although depression is common among older people, it is often underdiagnosed and undertreated in primary care settings. Depressive symptoms are often overlooked and untreated because they co-occur with other common chronic health problems among older adults [1]. Screening and early detection of depressive symptoms among older people are critically important for appropriate interventions, care, and treatment.

Several studies were conducted in Vietnam to explore the prevalence of depression and associated factors among older people [5–8]. Most studies employed a moderate or small and unrepresentative sample of older people. Five studies were conducted with a sample of older people aged 60 and over ranging from 110 to 523 in one or two rural districts. These studies used Geriatric Depression Scales 4 items (GDS-4), 15 items (GDS-15), 30 items (GDS-30), and the Patient Health Questionnaire 9 items (PHQ-9) to assess depressive symptoms [5–8, 11]. Only two studies were conducted with a large and representative sample of older people, but they did not use a scale specifically designed to assess depressive symptoms among older people [9, 10]. One study conducted with 6,050 people aged 60 and over from 12 provinces of Vietnam used the brief 11-item Epidemiological Studies-Depression scale (CES-D) to assess depressive symptoms [9]. The other study was conducted with 4,007 people aged 60 and over, sampled from 12 provinces. This study used one question asking about older people's feelings of sadness or depression to define depression status [10].

To the best of our knowledge, there have been no studies conducted in Vietnam with a large and representative sample of older people that used a screening tool that is specifically

designed to screen depression symptoms among older people. Our study analyzed nationally representative data on older people from the Vietnam National Aging Survey (VNAS) and used GDS-15, which is a scale specifically designed to assess depressive symptoms for older people. The study aims to explore the prevalence of depression and associated factors among Vietnamese older people.

## Materials and methods

### Data source

This study analyzed data from the Vietnam National Aging Survey (VNAS) conducted with a nationally representative sample of 3,006 older people aged 60 and over from 12 provinces of Vietnam. The recruitment of older people aged 60 and over and data collection were conducted from 08/01/2022 to 10/30/2022. The data were accessed for this research on 05/12/2023.

The VNAS sample was distributed proportionately to the older population (those aged 60 and over) sizes in six socio-economic regions of Vietnam including both urban and rural areas. This study employed a cross-sectional survey with a multiple-stage sampling method. Twelve provinces were selected using a population proportionate to size (PPS) method. In each selected province, 3 districts were selected using PPS. Four communes (the lowest administrative unit in Vietnam) per district were selected using PPS. In each selected commune, 2 villages were selected using a systematic random sampling method. In each selected village, 11 people (of which 4 people aged 60–69, 4 people aged 70–79, and 3 people aged 80 and over) were selected following a systematic random method. A sample replacement was implemented when the selected older person had moved to another place, was away from home at the time of data collection, and/or refused to be interviewed. In these cases, the subject was replaced with the next older person in the sampling list. With a response rate of 97.3%, a total sample of 3,183 older people aged 60 and over were selected for VNAS. Because 177 older people who were too weak and unable to answer the GDS, they were excluded from the analysis for this study. A total of 3,006 older people were finally included in the data analysis for this study.

In-person interviews were employed following a structured Vietnamese questionnaire on the tablet using the Census and Survey Processing System (CSPro), a computer-assisted personal interviewing program (CAPI) [18]. Twenty-four persons who were last-year students or recent graduates from public health, sociology, or social science Universities were trained to be interviewers. Six field supervisors were responsible for quality control of the data collection. Field supervisors observed selected interviews for each interviewer to ensure that the interviewer followed data collection and interviewing procedures. The CSpro data entry form was built with interactive logic and consistency checks. The program would show data errors immediately on the tablet screen during the interview so the interviewer could check and correct the errors in real-time. Field supervisors checked all completed questionnaires at the end of the data collection day by running a data-checking program. Field supervisors then asked interviewers to correct errors and fill in any missing information.

The institutional review boards of the Institute of Social Medical Studies approved the VNAS (Decision 04/HDDD-ISMS, dated 18 July 2022). Written informed consent was obtained from all respondents who participated in the study.

### Measures

**Dependent variable.** The 15-item Geriatric Depression Scale (GDS-15) [19] was used to assess depressive symptoms. The Vietnamese version of GDS-15 was validated and used in

several studies with older people in Vietnam [8, 11, 20]. In this study, the internal consistency of GDS-15 was good with a Cronbach's alpha of 0.79.

Respondents were asked to respond "Yes" or "No" to each of all 15 questions.

These responses resulted in a total score for each respondent ranging from 0 to 15. A score of 6 and over identifies people with depressive symptoms. Scores ranging from 0–5 are classified as within normal limits, 6–9 as mild depression, and 10–15 as moderate-to-severe depression [21, 22].

**Independent variables.** Sociodemographic and economic variables included gender, age, marital status, educational status, ethnicity, living areas, wealth quintiles, employment in the past 12 months, having health insurance.

Wealth index scores were created from 8 questions about housing characteristics, household utilities, and household assets using principal component analysis (PCA), to measure the economic status of older people. The wealth index scores were categorized into wealth quintiles including 5 levels: the Poorest, Poor, Medium, Wealthy, and the Wealthiest.

Health status was measured using a self-rated health question asking respondents to assess their health status including 5 = "Very Poor", 4 = "Poor", 3 = "Fair", 2 = "Good", and 1 = "Very Good". In data, analysis, to be comparable to other previous studies, (see, for instance, [9, 23–26]) the responses were grouped to: Very Poor/Poor, Fair, and Good/Very Good.

Having chronic diseases was assessed using one question asking if the respondent has ever been diagnosed with chronic diseases including Arthritis, Angina, Diabetes, Chronic Lung diseases (emphysema, bronchitis, chronic obstructive pulmonary diseases), Blood pressure problems, Cancer, Kidney diseases, Heart diseases, Liver diseases. A participant was defined as having "chronic disease" if she/he reported being diagnosed with any of the above diseases.

Activities of Daily Living's (ADL) limitation: the Index of Independence in ADL [27] was used to indicate the degree of difficulty in performing 5 basic activities including eating, dressing, bathing, moving in and out of bed, going to the toilet without help in the last 30 days. A score of 0 means no difficulty and 1 means difficulty for each activity. A total score ranges from 0 to 5 with 0 indicating no difficulty or completely independent. "ADL limitation" was defined if the score was equal to or greater than 1. The ADL was validated and used in several studies with older people in Vietnam [10, 11, 28]. In this study, the internal consistency of the ADL was good with a Cronbach's alpha of 0.89.

The Lawton Instrumental Activities of Daily Living (IADL) Scale [29] was used to assess independent living skills in 8 domains of functions including the ability to use telephones; shopping; housekeeping; laundry; mode of transportation; responsibility for own medications; and ability to handle finances. A respondent's functional level for each activity in the domains was rated with 0 indicating "low or no function" and 1 indicating "good or high function". The total score ranges from 0 (low function, dependent) to 8 (high function, independent). "IADL limitation" was defined as the score was equal to or greater than 1. The IADL was validated and used in previous studies in Vietnam. In this study, the internal consistency of the IADL was good with a Cronbach's alpha of 0.90.

Domestic violence was assessed by three questions asking if the respondent experienced verbal violence, was refused to talk, hit, beaten, or threatened by family members in the last 12 months. Domestic violence was defined if the respondent responded "Yes" to any of the three questions.

Cigarette smoking status was assessed using two questions asking if the respondent currently smoked cigarettes and/or waterpipe and if the respondent has ever smoked 100 cigarettes and/or waterpipe. Older people were categorized as non-smokers, former smokers, or current smokers.

Alcohol drinking behavior was assessed using two questions asking if the respondent drank alcohol in the last 6 months and how often the respondent drank alcohol in the last 6 months. Frequencies of drinking included none, less than once a month, monthly, weekly, and daily drinking.

Social support was measured using 5 questions asking about living arrangements (e.g., living with children), number of family members, number of living children, receiving financial support for daily living in the last 12 months, and social participation in the last 12 months.

## Data analysis

VNAS data was weighted to accurately present the older population. Weighted VNAS data was considered nationally representative of Vietnamese older people. Data were analyzed using Stata (version 14.0). Descriptive statistics were used to summarize older people's characteristics and prevalence of depression. Bivariate analyses were used to explore the associations between depression and other independent variables. The association significance of categorical variables was assessed by chi-square tests, while the association significance of continuous variables was assessed using t-tests. Multivariate analyses using logistic regression were used to assess the associations between depression and other independent variables. Independent variables having a p-value < 0.2 in the bivariate analyses were included in the logistic regression models [30]. Multicollinearity was checked using Tolerance and VIF criteria to ensure that the independent variables included in the logistic regression models were not highly correlated with each other.

## Results

### Older people's sociodemographic characteristics and prevalence of depressive symptoms

A total of 3,006 older people aged 60 and over were included in the study, of which 41.9% were male and 58.1% were female. The average older people's age was 69.8 (±8.2). About 66% were married and 34% were single, separated, divorced, or widowed. In terms of ethnicity, 95.8% were Kinh ethnic, which is the majority ethnicity in Vietnam. Regarding educational level, 20.3% had a high school education and above. In terms of productive activities, 43.1% were currently employed. Regarding wealth, 15.8% were in the poorest quintile and 28.5% were in the wealthiest quintile. By place of residence, 67.1% lived in rural areas and 32.9% lived in urban areas. For health insurance, 97.4% had at least one type of health insurance. Regarding living arrangements and social support, 87.8% lived with a spouse or children, 82.4% participated in social events in the last 12 months, and 84.6% received financial support for daily living. In terms of health status, 40.6% reported very poor or poor health status, 42.3% had ADL limitations, 32.2% had IADL limitations, and 78.3% had chronic diseases. Regarding health behavior, 17.9% were current smokers, and 8.6% drank alcohol daily. About 9% of older people experienced domestic violence in the last 12 months (Table 1).

The prevalence of depressive symptoms (GDS-15 score equal to and greater than 6) was 20.2%, in which 14.3% had mild depression, and 5.9% had moderate/severe depression (Table 1).

### Factors associated with depressive symptoms among older people

Table 2 shows the results of bivariate analyses examining the correlation between depressive symptoms and other factors. Gender, age, education, marital status, place of residence, wealth quintiles, employment, living arrangements, number of family members, number of children,

**Table 1. Characteristics of older people.**

| Characteristics (N = 3,006) | Categories | Total (%) |
|---|---|---|
| **Gender** | Male | 41.9 |
| | Female | 58.1 |
| **Age** | 60–69 | 58.3 |
| | 70–79 | 26.8 |
| | 80+ | 14.9 |
| | Mean±SD | 69.8±8.2 |
| **Education** | No schooling/Incomplete primary education | 34.4 |
| | Primary school | 18.6 |
| | Secondary school | 26.7 |
| | High school and above | 20.3 |
| **Marital status** | Single/Never married/Separated/Divorced/Widowed | 34.1 |
| | Currently married | 65.9 |
| **Place of residence** | Urban | 32.9 |
| | Rural | 67.1 |
| **Ethnicity** | Other | 4.2 |
| | Kinh | 95.8 |
| **Wealth quintiles** | Poorest | 15.8 |
| | Poor | 17.2 |
| | Medium | 18.7 |
| | Wealthy | 19.8 |
| | Wealthiest | 28.5 |
| **Employment** | Currently not employed | 56.9 |
| | Currently employed | 43.1 |
| **Having health insurance** | No | 2.6 |
| | Yes | 97.4 |
| **Depression** | No/Normal (0–5) | 79.8 |
| | Yes (6–15) | 20.2 |
| | Mild depression (6–9) | 14.3 |
| | Moderate to Severe depression (10–15) | 5.9 |
| **Living arrangements** | Live alone | 7.9 |
| | Live with a spouse or children | 87.8 |
| | Live with others | 4.3 |
| **Number of family members** | Mean±SD | 4.10±2.28 |
| **Number of living children** | Mean±SD | 6.86±4.06 |
| **Social participation** | No | 17.6 |
| | Yes | 82.4 |
| **Receiving financial support for daily living** | No | 15.4 |
| | Yes | 84.6 |
| **Self-rated health status** | Good/Very Good | 12.1 |
| | Fair | 47.3 |
| | Very poor/Poor | 40.6 |
| **ADL limitations** | No | 57.7 |
| | Yes | 42.3 |
| | Mean±SD | 0.96±1.49 |
| **IADL limitations** | No | 67.8 |
| | Yes | 32.2 |
| | Mean±SD | 0.86±1.68 |

(*Continued*)

**Table 1.** (Continued)

| Characteristics (N = 3,006) | Categories | Total (%) |
| --- | --- | --- |
| Have chronic diseases | No | 21.7 |
| | Yes | 78.3 |
| | Mean±SD | 1.72±1.45 |
| Smoking status | Non-smoker | 63.1 |
| | Former smoker | 19.0 |
| | Current smoker | 17.9 |
| Frequency of drinking | None | 68.9 |
| | Less than once a month | 8.5 |
| | Monthly | 7.2 |
| | Weekly | 6.8 |
| | Daily | 8.6 |
| | Drinking frequency score (range 1–5) | 1.78±1.33 |
| Domestic violence | No | 91.0 |
| | Yes | 8.9 |

social participation, self-reported health status, ADL/IADL limitations, having chronic diseases, cigarette smoking status, alcoholic drinking frequency, and experiencing domestic violence in the last 12 months were significantly associated with depressive symptoms ($p < 0.05$).

The prevalence of depressive symptoms was higher among female, unmarried, more advanced age, non-employed, lower education level, lower wealth quintiles, and rural older people compared with their male, married, younger, employed, higher education level, higher wealth quintiles, and urban counterparts.

Older people who lived alone and did not participate in social activities had a higher proportion of depressive symptoms than those who lived with spouse/children or lived with others, and participated in social activities, respectively. Older people who had depressive symptoms had fewer family members and more children than those who did not have depressive symptoms.

A higher prevalence of depressive symptoms was found among older people who never smoked cigarettes, who did not drink or drank less frequently in the last 6 months compared with those who smoked and who drank more frequently, respectively.

Older people who reported poor health status, who had ADL or IADL limitations, and who had chronic diseases had a higher proportion of depressive symptoms than those who reported fair or good/very good health status, who did not have ADL or IADL limitations, and who did not have chronic diseases, respectively.

The prevalence of depressive symptoms among older people who experienced domestic violence in the last 12 months was much higher than among those who did not experience domestic violence.

Results from multivariate analyses presented in Table 2 show that gender, place of residence, wealth quintiles, self-reported health status, ADL/IADL limitations, and experiencing domestic violence were significantly associated with depressive symptoms among older people.

Female older people were 2.2 times more likely than male older people to have depressive symptoms (OR = 2.21, 95% CI 1.34–3.62).

Rural older people were 1.8 times more likely than their urban counterparts to have depressive symptoms (OR = 1.83, 95% CI 1.15–2.89).

**Table 2. Factors associated with depressive symptoms among older people.**

| Characteristics (N = 3,006) | Categories | Depressive symptoms | | | | Adjusted OR (95% CI) |
|---|---|---|---|---|---|---|
| | | *No* | | *Yes* | | |
| | | *n* | *%/ Mean±SD* | *n* | *% /Mean±SD* | |
| **Gender** | Male *(ref.)* | 1041 | 87.8 | 193 | 12.2 | |
| | Female | 1249 | 73.9 | 523 | 26.1*** | 2.21 (1.34–3.62)** |
| **Age** | 60–69 | 914 | 83.4 | 231 | 16.6*** | |
| | 70–79 | 847 | 77.6 | 273 | 22.4 | |
| | 80+ | 529 | 69.4 | 212 | 30.6 | |
| | (Mean±SD) | 2290 | 69.3±7.65 | 716 | 72.0±9.9*** | 0.99 (0.97–1.01) |
| **Education** | No schooling/Incomplete primary education *(ref.)* | 897 | 70.7 | 400 | 29.3*** | |
| | Primary school | 460 | 78.9 | 138 | 21.1 | 1.08 (0.75–1.54) |
| | Secondary school | 593 | 83.4 | 129 | 16.6 | 1.06 (0.71–1.56) |
| | High school and above | 340 | 91.2 | 49 | 8.8 | 0.81 (0.45–1.43) |
| **Marital status** | Currently married *(ref.)* | 1451 | 84.7 | 330 | 15.3 | |
| | Single/Never married/Separated/Divorced/Widowed | 839 | 70.2 | 386 | 29.8*** | 1.49 (0.97–2.29) |
| **Place of residence** | Urban *(ref.)* | 416 | 88.0 | 81 | 12.0*** | |
| | Rural | 1874 | 75.7 | 635 | 24.3 | 1.83 (1.15–2.89)* |
| **Ethnicity** | Other *(ref.)* | 282 | 75.4 | 116 | 24.6 | |
| | Kinh | 2008 | 79.9 | 600 | 20.1 | |
| **Wealth quintiles** | Wealthiest *(ref.)* | 526 | 90.3 | 68 | 9.7 | |
| | Wealthy | 506 | 86.5 | 97 | 13.5 | 0.89 (0.52–1.53) |
| | Medium | 477 | 78.6 | 134 | 21.4 | 1.51 (0.93–2.46) |
| | Poor | 422 | 72.7 | 175 | 27.3 | 1.51 (0.88–2.59) |
| | Poorest | 359 | 61.4 | 242 | 38.6*** | 2.26 (1.39–3.66)** |
| **Employment** | Currently not employed *(ref.)* | 1318 | 75.8 | 515 | 24.2** | |
| | Currently employed | 972 | 85.0 | 201 | 15.0 | 0.86 (0.57–1.29) |
| **Having a health insurance** | No | 96 | 77.1 | 29 | 22.9 | |
| | Yes | 2194 | 79.8 | 687 | 20.2 | |
| **Living arrangements** | Alone *(ref.)* | 177 | 60 | 114 | 40*** | |
| | Live with spouse or children | 2056 | 81.9 | 567 | 18.1 | 0.58 (0.32–1.05) |
| | Others | 57 | 73.2 | 35 | 26.8 | 1.04 (0.38–2.82) |
| **Number of family members** | Mean±SD | 2290 | 4.2±2.24 | 716 | 3.7±2.36** | 0.98 (0.91–1.06) |
| **Number of living children** | Mean±SD | 2290 | 6.7±3.89 | 716 | 7.4±4.64** | 0.98 (0.95–1.02) |
| **Social participation** | No *(ref.)* | 375 | 65.7 | 235 | 34.3*** | |
| | Yes | 1915 | 82.8 | 481 | 17.2 | 0.78 (0.54–1.12) |
| **Receiving financial support for daily living** | No | 322 | 81.92 | 80 | 18.08 | |
| | Yes | 1968 | 79.36 | 636 | 20.64 | |
| **Self- rated health status** | Good/Very Good *(ref.)* | 268 | 97.7 | 7 | 2.3 | |
| | Fair | 1157 | 88.4 | 180 | 11.6 | 4.58 (1.99–10.57)*** |
| | Very poor/Poor | 865 | 64.4 | 529 | 35.6*** | 11.68 (4.96–27.49)*** |
| **ADL limitations** | No *(ref.)* | 1361 | 89.8 | 190 | 10.2*** | |
| | Yes | 929 | 66.1 | 526 | 33.9 | 2.12 (1.51 2.99)*** |
| | Mean±SD | 2290 | 0.7±1.23 | 716 | 1.9±1.98*** | |
| **IADL limitations** | No *(ref.)* | 1447 | 86.9 | 266 | 13.1*** | |
| | Yes | 843 | 64.8 | 450 | 35.2 | 1.61 (1.16–2.25)** |
| | Mean±SD | 2290 | 0.6±1.37 | 716 | 1.8±2.44*** | |

*(Continued)*

**Table 2.** (*Continued*)

| Characteristics (N = 3,006) | Categories | Depressive symptoms | | | | Adjusted OR (95% CI) |
| | | No | | Yes | | |
| | | n | %/ Mean±SD | n | % /Mean±SD | |
|---|---|---|---|---|---|---|
| **Have chronic diseases** | No (*ref.*) | 538 | 86.4 | 114 | 13.6** | |
| | Yes | 1752 | 77.9 | 602 | 22.1 | 0.99 (0.61–1.60) |
| | Mean±SD | 2290 | 1.60±1.36 | 716 | 2.19±1.71*** | |
| **Smoking status** | Non-smoker (*ref.*) | 1394 | 76.5 | 494 | 23.4*** | |
| | Former smoker | 456 | 86 | 104 | 14 | 1.29 (0.73–2.28) |
| | Current smoker | 440 | 84.4 | 118 | 15.6 | 1.00 (0.59–1.71) |
| **Frequency of drinking** | None (*ref.*) | 1528 | 77.4 | 569 | 22.6** | |
| | Less than once a month | 202 | 80.7 | 49 | 19.3 | |
| | Monthly | 163 | 82.2 | 33 | 17.8 | |
| | Weekly | 141 | 84.1 | 32 | 15.9 | |
| | Daily | 256 | 92.7 | 33 | 7.3 | |
| | Drinking frequency score (range 1–5) | 2290 | 1.8±1.35 | 716 | 1.5±1.12** | 1.08 (0.93–1.25) |
| **Domestic violence** | No (*ref.*) | 2156 | 82.9 | 557 | 17.1*** | |
| | Yes | 134 | 47.9 | 159 | 52.1 | 6.66 (4.00–11.05)*** |

**Note**: Ref: reference group,

*** p-value <0.001,

** p-value <0.01,

*p-value <0.05

All variables with a P-value <0.2 in the bivariate analyses were included in the logistic regression models.

Older people who were in the poorest quintile were 2.3 times more likely to have depressive symptoms than those in the wealthiest quintile (OR = 2.26, 95% CI 1.39–3.66).

Older people who reported their health as very poor/poor or fair were 11.6 and 4.6 times more likely to have depression symptoms compared with those who reported their health as good/very good, respectively (OR = 11.68, 95% CI 4.96–27.49; OR = 4.58, 95% CI 1.99–10.57).

Older people who had ADL and IADL limitations were 2.1 and 1.6 times more likely to have depression symptoms compared with those without ADL and IADL limitations, respectively (OR = 2.12, 95% CI 1.51–2.99; OR = 1.61 95%, CI 1.16–2.25).

Older people who experienced domestic violence in the last 12 months were 6.6 times more likely to have depressive symptoms than those who did not experience domestic violence (OR = 6.66, 95% CI 4.00–11.05).

## Discussion

Using data from a nationally representative sample of Vietnamese older people, our study found a high prevalence of depressive symptoms (20.2%) among them. This prevalence, consistent with the previously reported rates from other studies in Vietnam (with the prevalence ranging from 17.2% to 39.6%) shows that depression is common among older people [5–11].

The prevalence of depressive symptoms among older people in our study is lower than the prevalence reported by two studies conducted in Vietnam with a large and representative sample of older people, but different from our study, those studies used different depression screening tools rather than GDS. One study, which was conducted with 4,962 older people in 2018 and used the CES-D 11 items, found the prevalence of depression among older people

was 31.3% [9]. The other study conducted with a sample of 2,798 older people in 2011 using one question asking about sad or depressed mood to define depression status found that 39.6% of older people had depressive symptoms [10].

Among studies in Vietnam, only three studies used GDS to assess depression among older people. Two studies that used GDS-15 reported the prevalence of depressive symptoms among older people was 17.2% and 28.7% [8, 11] and the other that used GDS-30 reported the prevalence was 25.5% [6]. Using the same screening tool, but different from our study, those studies had a smaller and un-representative sample of older people.

Our study used GDS -15, which is a validated depression screening tool that was specifically designed for older people with high sensitivity and specificity in detecting depressive symptoms among older people [21, 31, 32]. The study was conducted with a large and nationally representative sample of 3,006 older people recruited from 12 provinces in 6 socio-economic regions of Vietnam. With a large and representative sample and a validated screening tool, this study therefore provided a good estimation of the prevalence of depression among Vietnamese older people.

This study found that being a female, living in rural areas, having a poor economic status, having a poor health status, having ADL limitations, having IADL limitations, and experiencing domestic violence were factors significantly associated with an increased risk of depressive symptoms among older people.

The prevalence of depressive symptoms among older women was higher than that of older men. This finding is consistent with previous studies in Vietnam and other countries that female gender is a risk factor for depression among older people [6–9, 23, 24, 28, 33, 34]. Epidemiological evidence consistently shows that depression is more common among women than men [35] and this gender difference in depression exists in late life [36]. Social, psychological, and biological factors including genetic vulnerability, hormones related to reproductive function, internalization coping strategies, gender-specific roles in society, and life stress account for the higher risk of depression among women [37, 38] Apart from the biological vulnerabilities, stressors that increase dramatically in old age (such as widowhood/living alone, poor health/chronic illness, cognitive decline, financial strain/poverty, and caregiving) contribute to an increased risk of depression among older women [36].

Economic status was found to be an associated factor with depression among older people. Older people in the poorest or poor quintiles were more likely to have depression than those in the wealthy and the wealthiest quintiles. Previous studies find that low income/economic status [5, 7, 9, 39, 40], and financial difficulties [24, 41] are associated with depression among older people. Those at the poorest economic level might not be able to afford daily living, and health care expenses, and might have economic burdens. The poor older people, who often live in poor housing conditions, are more exposed to environmental stresses such as pollution, temperature extremes, and poor sleeping environments. Poor economic status is also associated with poor nutrition, poor physical health, and increased exposure to trauma, violence, and crime. These effects contribute to a higher risk for mental illnesses including depression [42].

This study also found a higher prevalence of depressive symptoms among rural older people compared with their urban counterparts, like other studies such as [9, 43]. Rural older people are more likely to have lower incomes, to live in poorer housing conditions, and to have lower access to healthcare services than urban older people. The lower economic status may explain the increased prevalence of depression among rural older people.

This study, consistent with previous studies [9, 23–26], found a significant association between self-reported health status and depressive symptoms. Depressive symptoms were more prevalent among those who perceived their health as very poor/poor than among those

who reported a fair or good/very good health status. The direction of this relationship is also not clear. Poor health status may contribute to an increase in the likelihood of depression among older people. Depression may also adversely influence older people's health status. Older people with depression may be more likely to perceive a poorer health status. However, the finding further highlights that self-reported poor health status can indicate an increased risk of depression for older people in primary care settings.

The study found a higher prevalence of depression among older people who had ADL or IADL limitations. This finding is consistent with the evidence from other studies that ADL and IADL limitations are associated with depression among older people [11, 24, 28, 44–47]. The functional limitations related to basic activities and essential living skills, often as consequences of chronic physical health make the older people more dependent on others for daily living and care, constrain them from performing social life activities, and result in limited social life. These may contribute to an increased risk of depression among older people. Intervention programs for older people should pay more attention to ADL/IADL disabilities to prevent the risk of depression.

The study found a strong association between domestic violence and depressive symptoms among older people. Older people who experienced domestic violence in the last 12 months were much more likely to have depression than those who did not experience domestic violence. This study adds evidence from previous studies that domestic violence is a risk factor for depression among older people [8, 10, 34, 43]. Domestic violence, common among older people [48], brings traumatic stress and this may explain the consequent depression. Older people experience domestic violence, often long-term violence suffer from stress, fear, and isolation, and these consequently lead to depression [49]. Prevention and reduction of domestic violence toward older people is critical to prevent and reduce depression among older people.

This study has some limitations. First, it analyzed data from VNAS with a cross-sectional design to identify the association of depressive symptoms with associated factors. It could not determine whether the association between depressive symptoms with a factor was a causal relationship. For example, ADL limitations and poor health status might contribute to depressive symptoms and vice versa. Second, because GDS-15 is a screening tool rather than a diagnostic instrument, this study could only assess the prevalence of depressive symptoms rather than the prevalence of diagnosed depression among older people.

## Conclusion

Depressive symptoms among older people is prevalent among Vietnamese older people. Female gender, living in rural areas, low wealth quintiles, poor health status, ADL/IADL limitations, and experiencing domestic violence are predictors of depressive symptoms among older people.

Mental health services for older people, especially in rural and poor-economic areas should be increased and strengthened. Depression screening for older people should be included in primary care settings for early identification and treatment of depression.

Older women, those with poor health status and functional limitations, and those with poor economic status should be targeted for depression screening during medical visits at primary care facilities and community-based depression screening programs.

Physical and psychological support from families and communities should be provided to older people with poor health status and functional limitations and older people with domestic violence to improve mental health and reduce the risk of depression of those older people.

## Supporting information

**S1 Dataset.**
(DTA)

## Author Contributions

**Conceptualization:** Nam Truong Nguyen.

**Data curation:** Trang Nguyen.

**Formal analysis:** Nam Truong Nguyen, Trang Nguyen, Thu Dai Bui, Long Thanh Giang.

**Investigation:** Nam Truong Nguyen, Trang Nguyen, Thu Dai Bui, Long Thanh Giang.

**Methodology:** Nam Truong Nguyen, Trang Nguyen, Long Thanh Giang.

**Resources:** Trang Nguyen.

**Validation:** Nam Truong Nguyen, Trang Nguyen, Thu Dai Bui, Long Thanh Giang.

**Writing – original draft:** Nam Truong Nguyen.

**Writing – review & editing:** Nam Truong Nguyen, Trang Nguyen, Thu Dai Bui, Long Thanh Giang.

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
