## [Decision Letter · Decision Letter 0]

25 Oct 2023

PONE-D-23-21192Depression and associated factors among older people in Vietnam: findings from a National Aging SurveyPLOS ONE

Dear Dr. Nguyen,

Thank you for submitting your manuscript to PLOS ONE. After careful consideration, we feel that it has merit but does not fully meet PLOS ONE’s publication criteria as it currently stands. Therefore, we invite you to submit a revised version of the manuscript that addresses the points raised during the review process.

We look forward to receiving your revised manuscript.

Kind regards,

Qi Yuan

Academic Editor

PLOS ONE

Journal Requirements:

"The study utilized the data source from the Vietnam National Aging Survey (VNAS) conducted in 2022 by the Institute of Social Medical Studies and funded by the Asian Development Bank."

"The authors received no specific funding for this work."

Additional Editor Comments:

Please address the reviewers' comments, especially the concerns on the validity of GDS-15.

Reviewers' comments:

Reviewer's Responses to Questions

**Comments to the Author**

1. Is the manuscript technically sound, and do the data support the conclusions?

Reviewer #1: Partly

Reviewer #2: Yes

2. Has the statistical analysis been performed appropriately and rigorously? 

Reviewer #1: Yes

Reviewer #2: Yes

3. Have the authors made all data underlying the findings in their manuscript fully available?

Reviewer #1: No

Reviewer #2: Yes

4. Is the manuscript presented in an intelligible fashion and written in standard English?

Reviewer #1: No

Reviewer #2: Yes

5. Review Comments to the Author

Reviewer #1: Dear Editor,

I appreciate the opportunity to review the manuscript titled 'Depression and Associated Factors Among Older People in Vietnam: Findings from a National Aging Survey.' This study utilizes the Vietnam National Aging Survey (VNAS) conducted in 2022 to assess the prevalence of depressive symptoms and associated factors among older people in Vietnam. The findings indicate a 20.2% prevalence of depressive symptoms among older people and identify various factors associated with depression. I would like to offer several suggestions to enhance the manuscript's quality.

Abstract:

1. Methods: The abstract lacks information about the 'factors' evaluated in the study. Adding a brief statement about these factors will enhance clarity for readers and help them understand what was assessed.

Introduction:

2. Line 59: What is the current status of population aging in Vietnam? What percentage of the population is considered older, etc.?

3. Line 65: It appears that there are already previous studies (references 4-10) quantifying the burden of depression among the population of interest. Therefore, a robust justification for the current study is needed.

4. The authors offer a general justification, noting that prior research (references 4-10) relied on samples of moderate size and limited representativeness, and utilized scales that were not tailor-made for assessing depressive symptoms among the older adults. Enhancing the manuscript's clarity with details about the specific samples and scales employed in each of these studies would help describe these methodological constraints. For instance, it would be valuable to understand the characteristics of the samples used in those studies, the scales utilized, and how the present methodology addresses these limitations.

5. Line 78: Author provide general ration that previous studies (reference 4-10) employed a moderate or small and unrepresentative sample as well as use of scale specifically not designed to assess the depressive symptoms among older people. A description for each study on these methodological limitations will add clarity to evaluate their methodological limittaions. For example, what samples and scales were used by each of these studies and how does use of current tool address those limittaions?

6. Line 84-86: These statements seem more like methodological details. Including this level of methodological detail here may make the following paragraph sound repetitive.

Methods:

7. Is the VNAS dataset publicly available?

8. Could you please explain the terms "ecological regions" and "communes" in the local context? Adding explanatory phrases for such terms, which may not be familiar to international readers, would be beneficial.

9. Line 106: Please cite the 'Census and Survey Processing System,' the computer-assisted interview program.

10. Who collected the data, and how was the quality of data collection maintained? How many surveyors were involved?

11. What was the reliability of the GDS scale in the sample? Has the validity of the GDS been established in Vietnam? The same comment applies to other scales used in this study, such as IADL.

12. How was the wealth index score calculated from the described variables? Was it a simple addition? How many items were included, and what was the methodology?

13. On page 127, the authors mentioned, "In data analysis, the responses were grouped into: Very Poor/Poor, Fair, and Good/Excellent," but did not provide a rationale for combining these groups together.

14. On page 128, how was the disease information captured? Was it based on self-report, medical records, or some lab measures? Please provide specific details about the questions asked or measurements taken.

15. The measurement of variables like cigarette smoking and alcohol use is not clear and could benefit from a brief description of the specific questions asked.

16. How was social support measured? Was it a single-item question, or were pre-validated tools used?

17. Line 160: The definition of depression is already specified in the dependent variable section, so mentioning it here again seems redundant.

18. Regarding data analysis, how were the normality of variables assessed, and was multicollinearity checked?

19. Which variables were adjusted in the model?

Discussion:

20. What are the strengths of this study?

21. What are the implications of the findings for public health practice and policy? Please provide specific recommendations based on your findings.

Tables:

22. Please specify whether the OR in Table 2 is adjusted or unadjusted. Also, mention in the footnote the variables that were adjusted for.

Overall:

23. Grammar errors are prevalent in several parts of the manuscript, indicating the need for substantial editing and thorough proofreading to rectify grammar-related issues.

Thank you for considering these suggestions to enhance the manuscript’s clarity and rigor.

Reviewer #2: This study has examined the prevalence and risk factors of depressive symptoms among older adults in Vietnam

The strength of the study is the use of nationally representative data and the results suggest the role of chronic diseases and functional limitations on depressive symptoms. Several studies have identified the similar results across the globe.

Few comments

• In line number 106, you can mention it as Computer-assisted personal interviewing (CAPI) instead of computer assisted interview program

• What is the response rate of the survey?

• Validity of the 15-item Geriatric Depression Scale in Vietnam?, add the The Cronbach’s alpha value

• Whether the questions were translated in local language?

• You have included depression as a chronic and independent variable, there may some collinearity, therefore remove depression from the list of chronic diseases included as an independent variable

• In table 1, the prevalence of chronic diseases were not included except depression and pls remove depression. The list of variable included in the table 2 must be available in table 1 also

6. PLOS authors have the option to publish the peer review history of their article (what does this mean?). If published, this will include your full peer review and any attached files.

Reviewer #1: No

Reviewer #2: No

---

## [Author Response · Author response to Decision Letter 0]

7 Nov 2023

Response to Reviewers and Editor for the manuscript “Depression and associated factors among older people in Vietnam: findings from a National Aging Survey”

Dear PLOS ONE Editor, Reviewers,

We would like to express our gratitude to the editor and reviewers for the feedback and helpful comments to improve our manuscript.

We have made all attempts to fully address the editor and reviewers’ comments in the revised manuscript. We believe the additional revisions based on reviewers’ comments have helped to substantially improve our manuscript. The revised manuscript has been submitted alongside this response to the reviewer’s letter.

Below, we have outlined how we have handled the Editor’s additional requirements and reviewers’ comments. We have highlighted our responses to each comment in bold.

Responses to Journal Requirements

Response: We have revised and formatted the manuscript following PLOS ONE's style requirements.

"The study utilized the data source from the Vietnam National Aging Survey (VNAS) conducted in 2022 by the Institute of Social Medical Studies and funded by the Asian Development Bank."

"The authors received no specific funding for this work."

Response: 

We have deleted the statement about the funding of the Vietnam National Aging Survey (VNAS), which we analyzed data from for this paper in the Acknowledgement section. As we did not receive any funding for our work, we would like to keep our current Funding Statement "The authors received no specific funding for this work". 

Response: Following the journal request, we have made the data fully available.

We have uploaded a minimal anonymized data set as a Supporting Information file.

Additional Editor Comments:

Please address the reviewers' comments, especially the concerns on the validity of GDS-15.

Response: We have addressed all reviewers' comments, including the concerns about the validity of GDS-15.

Responses to Reviewers’ comments

Thank you very much for your valuable comments/suggestions. We have revised the manuscripts based on your comments/suggestions. Our responses to each of your comments are below.

We have highlighted our responses to each comment in bold.

Comments to the Author

1. Is the manuscript technically sound, and do the data support the conclusions?

Reviewer #1: Partly

Reviewer #2: Yes

Response: We have revised the method section and provided more details about the methodology following the reviewer’s comments. We hope that the revised manuscript is technically sound, and the data support the conclusions.

2. Has the statistical analysis been performed appropriately and rigorously? 

Reviewer #1: Yes

Reviewer #2: Yes

3. Have the authors made all data underlying the findings in their manuscript fully available?

Reviewer #1: No

Reviewer #2: Yes

Response: Following the reviewer’s comment and journal request, we have made the data fully available.

We have submitted a data set as a Supporting Information file.

4. Is the manuscript presented in an intelligible fashion and written in standard English?

Reviewer #1: No

Reviewer #2: Yes

Response: We have done a thorough proofreading and have made grammar corrections for the revised manuscript.

5. Review Comments to the Author

Responses to reviewer # 1’s comments

Reviewer #1: Dear Editor,

I appreciate the opportunity to review the manuscript titled 'Depression and Associated Factors Among Older People in Vietnam: Findings from a National Aging Survey.' This study utilizes the Vietnam National Aging Survey (VNAS) conducted in 2022 to assess the prevalence of depressive symptoms and associated factors among older people in Vietnam. The findings indicate a 20.2% prevalence of depressive symptoms among older people and identify various factors associated with depression. I would like to offer several suggestions to enhance the manuscript's quality.

Abstract:

1. Methods: The abstract lacks information about the 'factors' evaluated in the study. Adding a brief statement about these factors will enhance clarity for readers and help them understand what was assessed.

Response: We have added a list of factors associated with depression that the study assessed in the abstract/methods (lines 38-41).

Introduction:

2. Line 59: What is the current status of population aging in Vietnam? What percentage of the population is considered older, etc.?

Response: We have added the information “The number of people aged 60 and over was 12.58 million which accounted for 12.8% of the total population in 2021” (Lines 58-59).

3. Line 65: It appears that there are already previous studies (references 4-10) quantifying the burden of depression among the population of interest. Therefore, a robust justification for the current study is needed.

Response: We have added a justification for the need of the current study. (Lines 91-95 in the revised manuscript).

4. The authors offer a general justification, noting that prior research (references 4-10) relied on samples of moderate size and limited representativeness, and utilized scales that were not tailor-made for assessing depressive symptoms among the older adults. Enhancing the manuscript's clarity with details about the specific samples and scales employed in each of these studies would help describe these methodological constraints. For instance, it would be valuable to understand the characteristics of the samples used in those studies, the scales utilized, and how the present methodology addresses these limitations.

Response: We have added a description with details about the specific samples and scales employed in each of the prior studies conducted in Vietnam. (Lines 78-90 in the revised manuscript).

5. Line 78: Author provide general ration that previous studies (reference 4-10) employed a moderate or small and unrepresentative sample as well as use of scale specifically not designed to assess the depressive symptoms among older people. A description for each study on these methodological limitations will add clarity to evaluate their methodological limitations. For example, what samples and scales were used by each of these studies and how does use of current tool address those limitations?

Response: We have added a description with details about the specific samples and scales employed in each of the prior studies. (Line 78-90 in the revised manuscript). We have also added a justification of advantages of the current study with a large and representative sample of older people and the employment of GDS-15, a depression screening scale specifically designed for older people. (Lines 91-95 in the revised manuscript).

6. Line 84-86: These statements seem more like methodological details. Including this level of methodological detail here may make the following paragraph sound repetitive.

Response: We have deleted this statement and revised the purpose of the study (Lines 95-96 in the revised manuscript).

Methods:

7. Is the VNAS dataset publicly available?

Response: Following the reviewers’ comment and journal request, we have made the data fully available.

We have submitted a data set as a Supporting Information file.

8. Could you please explain the terms "ecological regions" and "communes" in the local context? Adding explanatory phrases for such terms, which may not be familiar to international readers, would be beneficial.

Response: Vietnam is divided into 6 socio-economic regions. Commune is the lowest administrative unit.

We replaced the term ‘ecological’ with ‘socio-economic’ (Line 105). An explanatory phrase for the term “commune” was added (Line 108). 

9. Line 106: Please cite the 'Census and Survey Processing System,' the computer-assisted interview program.

Response: We have added a reference for the 'Census and Survey Processing System (CSpro), the computer-assisted personal interview program, and have cited it (lines 120-121).

10. Who collected the data, and how was the quality of data collection maintained? How many surveyors were involved?

Response: Twenty-four persons who were last-year students or recent graduates from public health, sociology, or social science Universities were trained to be interviewers. Six field supervisors were responsible for quality control of the data collection. Field supervisors observed selected interviews for each interviewer to ensure that the interviewer followed data collection and interviewing procedures. The CSpro data entry form was built with interactive logic and consistency checks. The program would show data errors immediately on the tablet screen during the interview so the interviewer could check and correct the errors in real-time. Field supervisors checked all completed questionnaires at the end of the data collection day by running a data-checking program. Field supervisors then asked interviewers to correct errors and fill in any missing information.

Descriptions of data collectors and quality controls were added (line 121-130)

11. What was the reliability of the GDS scale in the sample? Has the validity of the GDS been established in Vietnam? The same comment applies to other scales used in this study, such as IADL.

Response: The Vietnamese version of GDS-15 was validated and used in several studies with older people in Vietnam. In this study, the internal consistency of GDS-15 was good with a Cronbach’s alpha of 0.79.

We have added this information with citations (lines 137-139). 

The ADL and IADL were validated and used in several studies with older people in Vietnam. In this study, Cronbach’s alpha for the ADL was 0.89, and for the IADL was 0.9.

These information were added (lines 167-169, lines 176-178).

12. How was the wealth index score calculated from the described variables? Was it a simple addition? How many items were included, and what was the methodology?

Response: We used principal component analysis (PCA) to create a wealth index score from 8 questions about housing characteristics, household utilities, and household assets. 

We have added this description in the method section (lines 149-152).

13. On page 127, the authors mentioned, "In data analysis, the responses were grouped into: Very Poor/Poor, Fair, and Good/Excellent," but did not provide a rationale for combining these groups together.

Response: Many previous studies grouped self-reported health status into 3 categories. To be comparable with previous studies we combined the self-reported health status responses into 3 categories: Very Poor/Poor, Fair, and Good/Very Good.

We have added this rationale with citations (lines 155-156).

14. On page 128, how was the disease information captured? Was it based on self-report, medical records, or some lab measures? Please provide specific details about the questions asked or measurements taken.

Response: The disease information was captured based on self-report. 

We used one question asking if the respondent has ever been diagnosed with chronic diseases including Arthritis, Angina, Diabetes, Chronic Lung diseases (emphysema, bronchitis, Chronic obstructive pulmonary diseases), Blood pressure problems, Cancer, Kidney diseases, Heart diseases, Liver diseases.

We have added this information (lines 157-158).

15. The measurement of variables like cigarette smoking and alcohol use is not clear and could benefit from a brief description of the specific questions asked.

Response: We have added brief descriptions of the specific questions asking about cigarette smoking and alcohol use (lines 182-187).

16. How was social support measured? Was it a single-item question, or were pre-validated tools used?

Response: Social support was measured using 5 questions asking about living arrangements (e.g., living with children), number of family members, number of living children, receiving financial support for daily living in the last 12 months, and social participation in the last 12 months. 

We have added this information in the measure section (lines 188-190).

17. Line 160: The definition of depression is already specified in the dependent variable section, so mentioning it here again seems redundant.

Response: We deleted this definition of depression in the data analysis section.

18. Regarding da

---

## [Decision Letter · Decision Letter 1]

16 Feb 2024

Depression and associated factors among older people in Vietnam: findings from a National Aging Survey

PONE-D-23-21192R1

Dear Dr. Nguyen,

We’re pleased to inform you that your manuscript has been judged scientifically suitable for publication and will be formally accepted for publication once it meets all outstanding technical requirements.

Kind regards,

Qi Yuan

Academic Editor

PLOS ONE

Additional Editor Comments (optional):

Reviewers' comments:

Reviewer's Responses to Questions

**Comments to the Author**

1. If the authors have adequately addressed your comments raised in a previous round of review and you feel that this manuscript is now acceptable for publication, you may indicate that here to bypass the “Comments to the Author” section, enter your conflict of interest statement in the “Confidential to Editor” section, and submit your "Accept" recommendation.

Reviewer #2: All comments have been addressed

2. Is the manuscript technically sound, and do the data support the conclusions?

Reviewer #2: Yes

3. Has the statistical analysis been performed appropriately and rigorously? 

Reviewer #2: Yes

4. Have the authors made all data underlying the findings in their manuscript fully available?

Reviewer #2: Yes

5. Is the manuscript presented in an intelligible fashion and written in standard English?

Reviewer #2: Yes

6. Review Comments to the Author

Reviewer #2: The authors have addressed all the comments, i have few more comments

In the topic and other places, instead of older people, you can write older adults

In the topic, instead of depression, use depressive symptoms

In regression, authors can adjust for provinces

7. PLOS authors have the option to publish the peer review history of their article (what does this mean?). If published, this will include your full peer review and any attached files.

Reviewer #2: No

---

## [Editor Report · Acceptance letter]

29 Apr 2024

PONE-D-23-21192R1 

PLOS ONE

Dear Dr. Nguyen, 

I'm pleased to inform you that your manuscript has been deemed suitable for publication in PLOS ONE. Congratulations! Your manuscript is now being handed over to our production team.

Kind regards, 

on behalf of

Dr. Qi Yuan 

Academic Editor

PLOS ONE